# Diversity and Hydrocarbon-Degrading Potential of Deep-Sea Microbial Community from the Mid-Atlantic Ridge, South of the Azores (North Atlantic Ocean)

**DOI:** 10.3390/microorganisms9112389

**Published:** 2021-11-19

**Authors:** Maria Paola Tomasino, Mariana Aparício, Inês Ribeiro, Filipa Santos, Miguel Caetano, C. Marisa R. Almeida, Maria de Fátima Carvalho, Ana P. Mucha

**Affiliations:** 1CIIMAR—Interdisciplinary Centre of Marine and Environmental Research, University of Porto, Terminal de Cruzeiros do Porto de Leixões, Avenida General Norton de Matos, s/n, 4450-208 Matosinhos, Portugal; mariana_aparicio_30@hotmail.com (M.A.); inesfcribeiro@gmail.com (I.R.); pipa.santos2@gmail.com (F.S.); mcaetano@ipma.pt (M.C.); calmeida@ciimar.up.pt (C.M.R.A.); mcarvalho@ciimar.up.pt (M.d.F.C.); amucha@ciimar.up.pt (A.P.M.); 2Instituto Português do Mar e da Atmosfera, I.P. Avenida de Brasília, 1449-006 Lisboa, Portugal; 3School of Medicine and Biomedical Sciences (ICBAS), University of Porto, Rua de Jorge Viterbo Ferreira 228, 4050-313 Porto, Portugal; 4Faculty of Sciences, University of Porto, Rua do Campo Alegre 790, 4150-171 Porto, Portugal

**Keywords:** microbial consortia, petroleum hydrocarbons, bioremediation, deep-sea, 16S rRNA gene, next-generation sequencing

## Abstract

Deep-sea sediments (DSS) are one of the largest biotopes on Earth and host a surprisingly diverse microbial community. The harsh conditions of this cold environment lower the rate of natural attenuation, allowing the petroleum pollutants to persist for a long time in deep marine sediments raising problematic environmental concerns. The present work aims to contribute to the study of DSS microbial resources as biotechnological tools for bioremediation of petroleum hydrocarbon polluted environments. Four deep-sea sediment samples were collected in the Mid-Atlantic Ridge, south of the Azores (North Atlantic Ocean). Their autochthonous microbial diversity was investigated by 16S rRNA metabarcoding analysis. In addition, a total of 26 deep-sea bacteria strains with the ability to utilize crude oil as their sole carbon and energy source were isolated from the DSS samples. Eight of them were selected for a novel hydrocarbonoclastic-bacterial consortium and their potential to degrade petroleum hydrocarbons was tested in a bioremediation experiment. Bioaugmentation treatments (with inoculum pre-grown either in sodium acetate or petroleum) showed an increase in degradation of the hydrocarbons comparatively to natural attenuation. Our results provide new insights into deep-ocean oil spill bioremediation by applying DSS hydrocarbon-degrading consortium in lab-scale microcosm to simulate an oil spill in natural seawater.

## 1. Introduction

Petroleum consists of a complex mixture of diverse aliphatic and aromatic hydrocarbons, resins, and asphaltenes [1], which affects crude oil’s susceptibility to biodegradation and their environmental fate [2]. The high hydrophobicity and low solubility of hydrocarbon pollutants contribute to their accumulation in the marine sediments [3,4]. Oil contamination may have a greater impact on more sensitive environments, such as cold marine areas, where the harsh conditions lower the rate of natural attenuation, allowing the petroleum pollutants to persist more than 20 years in marine soil, permafrost, and deep-sea waters [5,6], raising problematic environmental concerns. 

Microbial-based bioremediation represents a sustainable and cost-effective strategy that accelerates the removal of environmental pollutants [7] by using efficient degrading remediators [8]. Hydrocarbon-degrading bacteria are microorganisms that are able to use hydrocarbons as energy sources for growth, although they also require other essential nutrients that could be limited in the ecosystems. Biostimulation, the addition of the adequate supplement of nitrogen (N) and phosphorus (P) to avoid metabolic limitations, allows optimizing the environmental conditions for microbial degradation to remove pollutants, enhancing the bioremediation efficiency. However, the scarcity of indigenous microbes with the proper metabolic predispositions to degrade petroleum hydrocarbons, could represent a limitation of this approach [9]. Bioaugmentation, through the addition of microorganisms with biodegradation/detoxification capacity [10] such as specific oil-degrading bacteria that could be previously isolated from a contaminated (or pre-contaminated) site, could be an option. The success of biotechnological solutions for bioremediation application depends on the identification and selection of the indigenous hydrocarbon-degrading microbial assemblages from a target contaminated environment and enhance their biodegradation potential. At the same time, the optimization of the lab-grown condition of the bacterial strains selected are needed to improve the efficiency of bioaugmentation approaches and bioremediation performance [10,11,12,13,14]. Therefore, even though bioremediation is seen as an efficient tool for removing pollutants, investigation in this field is still required to expand knowledge regarding microorganisms, native to different environmental niches, with a good remediation capacity and adaptability.

Marine cold regions represent some of the largest biotopes on the Earth, including deep-sea sediments (DSSs) that cover almost two thirds of the planet’s surface and host a surprisingly diverse microbial community [15]. DSSs, generally characterized by an oligotrophic environment with low temperatures, are also considered repository systems of hydrocarbons that fall through the water column to the ocean floor [16]. Thus, microorganisms residing in the DSSs, because of their adaptation to this extreme environment, have been shown to possess distinct strategies when naturally or accidentally exposed to several contaminants such as petroleum hydrocarbons [5]. 

Previous studies reported microbial prokaryotes and eukaryotes communities of putative oil degraders harbored in DSS and how they bloom in the presence of elevated hydrocarbons concentrations reporting their important role in natural bioremediation [17,18]. For this reason, the exploitation of DSS as a huge reservoir of microorganisms, both as single-cell strains as well as microbial consortia, can be a key issue for achieving pollutant bioremediation.

The recent development of high-throughput technologies, such as next generation sequencing (NGS), allowed to investigate and compare different deep-sea ecosystems providing information on their microbial diversity and helping to discover even novel taxa [15,19,20]. Different NGS-based surveys on deep-sea microbial communities have outlined the microbial responses and dynamics that occur over the spill [21]. However, it remains difficult to differentiate between environmental impacts and naturally occurring environmental changes that could drive the microbial dynamics [22], lacking a baseline characterisation of indigenous microbial diversity of the sample site. Furthermore, although many hydrocarbon degraders have been isolated from DSSs [16,23,24,25], the current knowledge about their potential for oil spill bioremediation is still limited. 

From this perspective, the present research aims to contribute to bring novelty in deep-ocean oil spill bioremediation by investigating the hydrocarbon-degrading potential of microorganisms isolated from a DSS (MAR, North Atlantic Ocean) and test it at lab scale in a microcosm-simulated oil spills. To this end, a combined approach was adopted: (1) culture-independent survey was used to explore the autochthonous deep-sea microbial diversity by next-generation sequencing of the V4–V5 hypervariable region of the 16S ribosomal RNA (rRNA) gene; (2) also, culture dependent techniques were applied, allowing to identify autochthonous deep-sea bacteria with bioremediation potential; a novel hydrocarbonoclastic DSS consortium of 8 bacterial strains, isolated after an enrichment process, was assembled. Then a microcosm bioremediation experiment was performed to test consortium hydrocarbon degradation potential in a context of bioaugmentation techniques with natural seawater and petroleum (detailed schematic description is reported in the workflow Appendix A).

## 2. Material and Methods

### 2.1. Sampling

Sampling was conducted during the oceanographic campaign EMEPC\PEPC\Luso\2016 in September 2016 on board of the Portuguese Navy Research Vessel NRP Almirante Gago Coutinho in the Mid-Atlantic Ridge, south of the Azores (North Atlantic Ocean).

Four surface sediment samples between −1065 m and −1073 m depth were collected in (location L16D04 described by Souto and Albuquerque 2019). Several dives were performed by the ROV ‘LUSO’ to collect three sediment samples (L1, L2 and L4) using a corer mini-corer (upper 5-cm depth) and an additional sample (composite sediment) was collected through suction of the surface of the different points (L3 composite sediment) during a ROV dive (Table 1).

For each location, two subsamples of deep-sea sediments were collected and stored in different conditions. One subsample was immediately stored at −80 °C for further genomic processing to unravel the microbial diversity in the deep sediment, while the other used to identify autochthonous hydrocarbon-degrading bacteria was preserved in sterile flask with lifeguard solution under refrigerated conditions, until processing in the laboratory.

### 2.2. Environmental DNA Extraction, Amplification and Sequencing

Environmental DNA (eDNA) was extracted from the four deep-sea sediment samples by the PowerSoil DNA Isolation Kit (QIAGEN) following the manufacturers protocol. Concentration of DNA extracted was measured using the Qubit fluorometric quantitation kit (Qubit dsDNA High Sensibility Assay Kit, Thermo Fisher Scientific Scientific Inc., Waltham, MA, USA).

Samples were then prepared and sent to Genoinseq (Cantanhede, Portugal) facilities for the amplification of the hypervariable V4-V5 region of the 16S rRNA gene using specific primers (forward 515F-Y, 5′-GTGYCAGCMGCCGCGGTAA-3′; and reverse 926R, 5′-CCGYCAATTYMTTTRAGTTT-3′) [26]. The amplified products obtained were purified and normalized with SequalPrep 96-well plate kit (ThermoFisher Scientific, Inc., Waltham, MA, USA) [27]. Pair-end sequencing was carried out in the Illumina MiSeq^®^ sequencer with the V3 chemistry (Illumina, Inc., San Diego, CA, USA) at Genoinseq laboratories. 

### 2.3. Bioinformatic Analysis

After sequencing, raw reads were extracted from Illumina MiSeq^®^System in fastq format and quality-filtered at Genoinseq facilities using default parameters [28,29]. FastQ reads were then converted into FASTA format and merged using Mothur software package (version 1.43.0) [30]. 

Silva Next Generation Sequencing pipeline (SilvaNGS) of the SILVA rRNA gene database project (Version: 1.9.5/1.4.6; SILVA: r138.1), was used for the taxonomic characterization of the prokaryotic community using default settings [31,32]. In brief, after quality controls, sequences were dereplicated and clustered into operational taxonomic units (OTUs) using the similarity threshold of 98% by cd-hit-est [33]; the classification was then performed against SILVA SSU Ref dataset release r138.1 using blastn version 2.2.30 + with standard settings [34,35]. Reads without any BLAST hits or reads with weak BLAST hits (%sequence identity + %alignment coverage/2 < 93), remained unclassified and were labelled as “no relative”. Rarefaction curves, number of OTUs, number of singletons, and Good’s coverage indices were also obtained inside SilvaNGS pipeline. Detailed protocol description is given in Bragança et al. [36]. Alpha-diversity indices (Shannon-Wiener Index, Simpson’s evenness Index) and richness estimator (Chao1) were calculated in R environment, using the R package vegan version 2.5-7. Raw sequences obtained in this study have been deposited in the SRA (NCBI) database with the accession number PRJEB39245.

### 2.4. Enrichment of Hydrocarbons-Degrading Bacteria from Deep-Sea Sediments 

At the laboratory, an enrichment experiment with crude oil was set up to recover autochthonous deep-sea bacteria with bioremediation potential to be used in a hydrocarbons-degrading bacterial consortium for further bioremediation experiments. For that, each sediment sample was placed in a flask and nutrient medium was added. The flasks were kept in agitation for 30 min at 30 rpm and then vortexed at max velocity for 1 min. One hundred microliters were transferred from each flask to a 100 mL serum flask and used as inoculum in 10 mL of Bushnell–Haas broth (Difco) supplemented with 2% NaCl (*v/v*), nutrients: N (added as KNO_3_ 4.04 g L^−1^) and P (added as KH_2_PO_4_ 1.08 g L^−^^1^), and 50 μL of crude oil (Arabian light crude-oil, provided by an oil refinery). The flasks were kept closed (to avoid contaminations), under constant agitation (100 rpm) favoring aerobic condition, at room temperature for 15 days. After this enrichment phase, bacteria present in the microbial culture were cultured in plate count agar (PCA) (Liofilchem, Roseto degli Abruzzi, Italy). Different colonies were described morphologically and purified. Isolated strains were then cryopreserved at −80 °C and biomass of each strain was also collected for phylogenetic identification. Bacterial DNA was extracted from all isolates with the kit E.Z.N.A. ^®^ Bacterial DNA (Omega Bio-tek). DNA quantification was performed with the kit Quant-it HsDNA in the Qubit fluorometer (Invitrogen). For phylogenetic identification, the full-length of the bacterial 16S rRNA gene was amplified with the universal primers 27F (5′-AGAGTTTGATCMTGGCTCAG-3′) and 1492R (5′-TACGGYTACCTTGTTACGACTT-3′) by Polymerase Chain Reaction (PCR).

The PCR mixture and PCR conditions were the same as Perdigão et al., 2021. Amplified samples were run in a 1.5% agarose gel containing SYBR Safe (Thermo Fisher Scientific, Waltham, MA, USA) and the resulting PCR products were sequenced at Genomics i3S Scientific Platform (Porto, Portugal). 16S rRNA sequences obtained were analyzed using the Geneious software (11.1.4) and the consensus sequences were identified using the NCBI BLAST database (https://blast.ncbi.nlm.nih.gov/Blast.cgi, accessed on 21 January 2021). The 16S rRNA gene sequences of the identified bacterial strains were deposited in GenBank (NCBI, Bethesda, MD, USA) under the accession numbers reported in Table 2.

### 2.5. Microcosm Bioremediation Experiment

Bacterial strains isolated from the L1 site were selected to assemble the Deep-Sea Sediment Hydrocarbons-Degrading Bacterial consortium (HDB_L1_DSS) to be used in the bioremediation experiment. For this, two inocula were prepared, one with petroleum and another with sodium acetate, based on a previous growth optimization experiment [37]. For inoculum preparation, a mixture of the eight bacterial strains were added to 250 mL sterilized glass flasks containing 20 mL BH medium supplemented with 2% NaCl, with an OD600 nm of 1. Half of the flasks were supplemented with petroleum (P) as the carbon source, in a 20:0.5 (*v*/*v*) ratio. The other half was supplemented with sodium acetate (A) that was added every day for four days to the cultures at a final concentration of 1 g L^−^^1^. For each carbon source, triplicate flasks were prepared and incubated closed for 4 days, at 28 °C, under constant agitation (100 rpm), favoring aerobic conditions. After this period, the cultures were centrifuged and the whole resulting pellet was re-suspended in 1 mL of natural seawater collected at Matosinhos beach (NW of Portugal), to create inoculum pre-grown in acetate (EA) or in petroleum (EP).

The bioremediation experiments were carried out in 250-mL flasks containing 20 mL of natural seawater and filtered crude oil added on a proportion of 20:0.5 (*v*/*v*). All the flasks were kept in darkness with constant agitation (100 rpm) under aerobic condition, at RT during 15 days. Four different treatments were tested: (i) natural attenuation (NA) (seawater + petroleum); (ii) biostimulation (BS) (seawater + petroleum + N&P nutrients); (iii) bioaugmentation (BAp) (seawater + petroleum + N&P nutrients + inoculum pre-grown in petroleum (EP)) and (iv) bioaugmentation (BA/a) (seawater + petroleum + nutrients + inoculum pre-grown in sodium acetate (EA)). As stated before, both biostimulation and bioaugmentation treatments were supplemented with nutrients, namely nitrogen (as KNO3) (4.04 g L^−1^ and phosphorus (as KH2PO4) (1.08 g L^−1^, to avoid N and P limitations and increase the metabolic activity. A defined percentage of N and P were calculated relative to the total oil mass on a weight ratio (C/N/P of 100:10:1). Natural seawater was selected as medium in the present experiment in order to simulate on lab-scale an oil spill in a natural environment in which different microbial communities normally interact. Natural attenuation was evaluated in flasks only with seawater and crude oil to estimate oil degradation by indigenous populations. BS, BAa and BAp were compared with NA to test the efficiency these bioremediation treatments. For natural attenuation (NA) treatment, six flasks were prepared. Three of them were immediately retrieved and placed at −20 °C to be considered T0 samples for later total petroleum hydrocarbons (TPHs) measurements. For the remaining treatments (BS, Bap, and BAa) three replicates per treatment were prepared. On the same day, triplicate most probable number method (MPN) tests in 96-well plates for all treatments were made as described in Section 2.6. This analysis was repeated on the last day of the bioremediation experiments (after 15 days). In addition, aliquots of both bioaugmentation treatments were diluted (10^−2^ to 10^−6^) in saline solution (8.5 g L^−1^) and spread in marine agar (MA), sea water agar (SW) and Bushnell-Haas (BH) as described later. Afterwards, the remaining solutions were frozen at −20 °C to stop microbial growth for later TPHs determination. 

### 2.6. Abundance of Hydrocarbon-Degrading Bacteria 

To assess the abundance of petroleum hydrocarbons degrading bacteria, a modified MPN protocol [38] was prepared in 96-well plates for all the treatments (NA, BS, BAa, BAp). 

Tenfold dilutions of the initial sample (20 μL) were inoculated in 180 μL of BH culture medium for microbial bacterial growth, supplemented with 2% NaCl, and 10 μL of filter sterilized crude oil (as the selective substrate for hydrocarbons degradation and the only carbon source available for microorganisms). After a 15 day-incubation period at room temperature, filtered iodonitrotetrazolium violet (INT) (3 g L^−^^1^) was added to each well (50 μL per well). Positive wells were scored after 24 h incubation at room temperature. To calculate MPN, a computer program was used (Mpncalc, Mike Curiale Calculator).

### 2.7. Determination of Total Petroleum Hydrocarbons (TPHs) 

For TPHs measurements, an adaption of the optimized protocol described by Couto et al. [39] was followed. For that, 5 mL of tetrachloroethylene (≥99% spectrophotometric grade, Sigma-Aldrich Steinhein, Germany) were added to each flask containing a sample. An ultrasonic (Elma, Transsonic 460/H model) extraction was carried out for 15 min and left resting for 10 min. The organic extract was then transferred to a second flask containing 0.3 g of hydrated (2%) silica gel (70–230 mesh, Macherey-Nagel), to remove non-mineral oil contaminants, and 1 g anhydrous sodium sulphate (Sigma-Aldrich). A second 15 min ultrasonic extraction was done by adding 5 mL of tetrachloroethylene to the initial flask containing the aqueous sample, followed by the same procedure. This second organic extract was mixed with the one from the first extraction. The extracts were then stirring for 10 min at 50 rpm (J.P Selecta, Unitronic-OR) and filtered through silanized glass wool (Sigma-Aldrich) in disposable pipette tips. Filtered extracts were analysed by Fourier transform infrared spectrophotometry (Jasco FT/IR-460 Plus) using a quartz cell of 1 cm path length (Starna Scientific Ltd, Essex, England). Calibration standards in tetrachloroethylene were prepared using a stock standard solution of equal volumes of isooctane (≥99% ACS spectrophotometric grade) and hexadecane (99%) solutions. TPHs were quantified by direct comparison with the calibration curve. The mean and respective standard deviation of three independent replicates per treatment was calculated. Results were expressed as percentage of degradation, considering TPHs measured in T0 samples and samples collected at the end of the experiment.

### 2.8. Isolation and Identification of Bacterial Strains of Microcosms’ Experiment

For both bioaugmentation (BAa and BAp) treatments tested in the bioremediation experiment, a combined sample of the respective triplicates was taken, ten-fold diluted in sterile saline solution (0.85%) and spread out onto three different agar media (MA, SW, and BH) to provide different nutritional conditions and to try to recover the bacterial strains used as inocula. Per litre, agar media were prepared with the following: MA with marine agar (55.2 g L^−1^) (Pronadisa); SW with 10 g soluble starch (Biochem Chemopharma), 4 g yeast extract (Liofilchem), 2 g peptone (Sigma-Aldrich), 33.3 g artificial sea salts (Sigma-Aldrich), 20 g agar (Liofilchem); BH with 3.27 g Bushnell-Haas broth (Difco™), 1 g sodium acetate anhydrous 99% (Alfa Aesar ^®^), 17.5 g agar (Liofilchem), 20 g NaCl (EMSURE ^®^). After 3–4 days of incubation at 28 °C, morphologically different colonies were described and purified. Isolated strains were then preserved in 21% glycerol at −80 C and biomass of each strain was also collected for DNA extraction. The 16S rRNA gene sequences of the bacterial identified strains were deposited in GenBank (Table 2).

### 2.9. Statistical Analysis and Data Visualization

Mean values (*n* = 3) and respective standard deviations of MPN and TPHs concentrations were determined per treatment. Significant relationships were considered when *p* values were below 0.05 (wilcoxon test *p* value). BarPlots were obtained in the R environment (version 3.2.2. Copyright 2015 The R Foundation for Statistical Computing), using base R and the “ggplot2” package. 

Venn diagrams were prepared as described by Bardou et al. [40] to describe the distribution of the low abundant/rare taxa (with abundance <1%) in the DSS environmental community. 

Phylogenetic tree of the isolated strains was made in order to complement the taxonomic studies. For this, an alignment was performed using the MUSCLE of the Geneious software with all the sequences and their three closest neighbor sequences in GenBank. The MEGA X: Molecular Evolutionary Genetics Analysis software was used to generate a Maximum Likelihood phylogenetic tree of 1412 bp with 1000 bootstraps based on the Tamura-Nei model [41]. The annotation and management of phylogenomic trees was performed by using iTOL v5 (http://itol.embl.de, accessed on 20 May 2021).

## 3. Results

### 3.1. Characterization of Deep Sediment Prokaryotic Community 

A total of 239,407 sequences of 16S rRNA gene were generated from the four DSS samples processed by the NGS analysis pipeline at the SILVA rRNA gene database project. Of these, 22,393 reads (c.a 9.3% of the total dataset) were rejected at the initial quality filtering step. In all 214,415 reads (98.56% of the total dataset) were classified and the remaining 2599 reads (c.a 1.08% of the total dataset) with percentage similarity to the closest relative below 93 in BLAST analysis were classified as ‘No relative’ reads (without any close relatives) (Appendix A).

All sequence libraries were quite far from saturation as shown by the rarefaction curves and good coverage indicating that a higher number of sequences is apparently required to cover the whole community diversity. Shannon diversity indices were very similar among the four samples (between 4.38 and 4.53), as well as Simpson’s evenness (average value 0.025); while chao indexes identified L3 as the sample with the highest diversity, in which was also recorded the highest number of OTUs and singletons (Appendix A). 

Taxonomic profile of the DSS prokaryotic community was performed at different taxonomic levels. Examining the relative abundance at the phylum level of the DSS community (Figure 1), all four samples were dominated by Proteobacteria (34.4 ± 3.5%). The second major phylum recorded in all the four samples was Planctomycetes (13.2 ± 5%), followed by Acidobacteria (12.3 ± 3.2%) and Thaumarchaeota (11.3 ± 2.2%). Lower percentages were found in Actinobacteria (3.6 ± 1.2%) and Firmicutes (3.5 ± 3.2%). Relatively to class level, five classes represented more than 50% of those with abundance >1%. Three of those classes belong to the Proteobacteria phylum (classes Gammaproteobacteria, Alphaproteobacteria and Deltaproteobacteria). Within the Proteobacteria, the Gamma class was the most abundant (17.5 ± 2.5% of total community), followed by the Alpha (10.4 ± 0.5%), and Delta (7.2 ± 1%) classes. The other two classes belong to Thaumarchaeota (class Nitrososphaeria (11.4 ± 2.2%) and to Planctomycetacia (Planctomycetes 6.2. ± 5.7%). 

When the analysis was made at lowest taxonomic level, the abundance of the top 15 taxa detected across the samples, represented around 40% of the community with abundance >1% (Appendix A). The average total number of taxa per sample was 653 ± 110, with the higher number detected in L3 (693) and the lowest in L2 (467). The most abundant taxon at lowest taxonomic level belongs to the Nitrosopumilaceae (average 9.1 ± 4.3%), followed by the Subgroup 10 (average 5.0 ± 3.8%) within the Thermoanaerobaculale, the NB1-j within the Deltaproteobacteria, Woeseia within the Gammaproteobacteria, Kiloniellaceae uncultured within Rhodovibrionales, Candidatus Nitrosopumilus genus within Nitrosopumilaceae family.

When we look at the microbial community at lowest level, we observed that it was represented by 96% of “low abundant” taxa with relative abundance below 1% (Appendix A) (2434 total taxa with an abundance <1% (L1 = 658, L2 = 443, L3 = 672, L4 = 661)/2530 total taxa (L1 = 682, L2 = 467, L3 = 698, L4 = 683)).

Zooming in this minor microbial population composed by taxa with lower abundance than 1% (Figure 2), we found that 245 (around 10%) of the low abundant taxa were common in all samples. The 245 common taxa included 4 of the 6 genera constituting HDB_L1_DSS isolated in this study after the enrichment process in the culture dependent analyses (see following result Section 3.2): *Pseudomonas* sp., *Acinetobacter* sp., *Microbacterium* sp., *Achromobacter* sp. (Figure 2), while the other two genera constituting the HDB_L1_DSS consortium were detected just in L1 and in two environmental samples (*Rheinheimera* sp. just in L1 with 0.03% and *Sphingobium* sp. in L1/L3 with 0.3% of relative abundance).

### 3.2. Taxonomic Identification of the Isolated Strains 

A total of 26 different bacterial strains were isolated from sediments of all sampling sites (L1–L2–L3–L4) and were identified using 16S rRNA gene sequence analysis. The sequences obtained were compared with the ones present in the NCBI database: 8 from L1, 5 from L2, 4 from L3, and 10 from L4 (see Table 2 for detailed description). The eight isolates from L1 were identified as *Achromobacter* sp., *Pseudomonas* sp., *Pseudomonas balearica*, *Rheinheimera aquimaris* and *Microbacterium testaceum*, *Acinetobacter* sp. and a *Sphingobium* sp. Among the four locations, L1 recorded the higher bacteria diversity (from this site have been recovered a total of 8 bacteria strains, four of them identified at species level, that belong to 6 different genera). For this reason, L1 isolates were selected to produce the HDB_L1_DSS to be used in the bioremediation experiments. 

Evolutionary relationships of the isolated strains can be observed in the 16S rRNA gene phylogenetic tree indicated in Figure 3. Different bioremediation treatments and culture media used for the isolation of hydrocarbon degrading microorganisms seem to have selected several strains belonging to ten different families, with the largest fraction of the strains belonging to the genus *Pseudomonas*. The phylogenetic analysis of the isolates attributed to the families Rhodobacteraceae and Flavobacteriaceae showed that the BPM3 and BPM4 strains formed a separate branch in the phylogenetic tree (Figure 3). The BPM3 and BPM4 strains belong to the *Aquicoccus* and *Frondibacter* genera, respectively, and present a similarity percentage of 96.97% and 95.71% with their closest neighbors of the GenBank (*Aquicoccus porphyridii strain* L1 8–17 for BPM3 and *Frondibacter mangrovi strain* 02OK1/10-76 for BPM 4) database. With these results, these strains can constitute a new taxon considering that the similarity limit for a new species is 98.7% [42].

### 3.3. Microcosm Bioremediation Experiment

In the microcosm experiment, the three bioremediation treatments, BS, BAp and BAa, were compared with natural attenuation (NA). Photos were taken for each treatment at the beginning and end of the experiment (after 15 days). Visual inspection of all flasks in the first day (T0) of the experiment showed a clear separation between the oil slick and the medium for the 4 treatments. At the end of the experiment (T15), it is easily noticeable that crude oil degradation occurred in all treatments except in the natural attenuation (NA), along with an increase in turbidity and petroleum’s droplets formed at the bottom and petroleum’s dispersion on the surface. These visual characteristics were more evident in BAp and BAa, suggesting a faster pace of biodegradation, which might be due to a higher microbial abundance since the beginning of the experiment (Figure 4c,d). This was also observed in the BS treatment even though it was less perceptible (Figure 4b).

Abundance of hydrocarbon degrading microorganisms was estimated at the beginning and at the end of the experiment. Results (Figure 5) showed that at the beginning of the experiment (T0) MPN results were below the detection limits for both natural attenuation (NA) and biostimulation. After 15 days of the experience, the abundance of hydrocarbon degraders increased, in agreement with the results observed by visual inspection of the flasks, confirming the microbial community had bioremediation potential and used crude oil as carbon source for growth. These results were much higher in the bioaugmentation (BAp and BAa) treatments, followed by biostimulation treatment, when compared with natural attenuation. However, MPN results indicate that the abundance of hydrocarbon degraders was already high in the first day of the experiment in the bioaugmentation (BAp and BAa MPN/mL >107) treatments, as observed by the saturation of the MPN methodology, so no difference could be truly noticed in these between the first and last days of the experiment. On the contrary, for N and BS treatments, results show an increase in the abundance of degraders during the 15 days of experiment.

Hydrocarbon degradation was estimated by comparing TPH concentrations at the beginning (T0) and at the end (T15) of the experiment. No significant differences were observed between treatments (Figure 6) although a tendency was observed for lower degradation in natural attenuation (18%), followed by biostimulation treatment (21%). The bioaugmentation with inoculum pre-grown in petroleum and bioaugmentation with inoculum pre-grown in acetate displayed the higher degradation rates, 23% and 30%, respectively.

### 3.4. Recover of Microorganisms from Microcosm Experiments

At the end of the microcosm experiment, samples of both bioaugmentation treatments (BAp and BAa) were spread in SW, MA, and BH media (in several dilutions 10^−2^ to 10^−6^) to understand which of the isolates used to assemble the HDB_L1_DSS were recovered at the end of the experiment. A total of 27 morphologically different strains were identified (Table 2). Not all the strains used to assemble the bioaugmentation consortium were recovered at the end of the bioremediation experiments. Beside the strains added in the initial HDB_L1_DSS consortium, other microorganisms were also recovered at the end of the experiment, which might have been originated from the seawater (collected at Matosinhos beach, Porto) used to assemble the microcosms, as they are described as marine bacteria.

## 4. Discussion

In this work, we were able to assemble a bacterial consortium (HDB_L1_DSS) with bacterial strains isolated from deep-sea sediment with potential for degrading petroleum hydrocarbons. This consortium was applied in a microcosm experiment with natural seawater and petroleum, in order to simulate an oil spill. Our results aim to contribute for the study of deep-sea microbial resources and for their exploitation in bioremediation of petroleum hydrocarbon polluted environments. 

Culture-independent approach was first adopted here to provide a baseline characterization of the microbial community residing in our four deep-sea sediments. NGS helped in shedding light on a great diversity of low abundant taxa, otherwise not reachable using culture dependent methods. The metabarcoding analysis of our four DSS environmental samples showed a similar microbial community composition, identifying a shared core structure, mainly composed of 4 phyla: Proteobacteria, Planctomycetes, Acidobacteria and Thaumarchaeota. Previous NGS studies focused on DSS, and reported that the abundance of these phyla can fluctuate depending on depth, organic carbon content, and geography [43,44]. The most abundant classes found in our environmental samples, Gammaproteobacteria and Alphaproteobacteria were also highlighted as the most representative in other surveys focused on deep cold marine sediments [16,45]. Different genera belonging to these classes were already described as key players in the biodegradation of oil compounds in DSS [23], indicating a potential for petroleum bioremediation of the microbial community of our sediments. A lower taxonomic characterization showed that the dominant group detected in the four environmental samples belongs to the family of Nitrosopumilaceae (within the phylum of Thaumarcheota and the order Nitrosopumilales) that has an important role as primary producers through ammonia oxidation [46]. Thaumarcheota group is ubiquitously distributed in deep-sea sediments, being particularly well adapted to the harsh and oligotrophic conditions that dominate these environments. It is interesting to highlight that some Thaumarchaeota were also described as degraders of low oil concentrations, but high hydrocarbons intrusions caused their disappearing in the contaminated environment since they can be rapidly outcompeted by other oil-degrading bacteria [47]. However, the role of Thaumarchaeota in oil degradation still remains poorly understood, since no direct evidence with cultured strains exists yet, and research on this topic is needed.

Despite the similar values of Shannon and Simpson’s evenness indices supported a general pattern of community heterogeneity among our DSS sample, rarefaction curves and Chao index point out a higher richness in the L3 sample. This result may be related with the different sediment nature of L3 that was collected through suction of the surface of the other 3 samples site. L3 may act as a sediment trap, therefore containing more substratum and harboring more OTUs than the other sites. Moreover, higher values of Chao index in L3 could be explained by the higher weight that Chao richness estimator gives to the low abundance species (singletons and doubletons) that were more abundant in L3 in comparison with the other sediment samples. Generally speaking, Chao represents an accurate richness estimator used to estimate the number of missing species that could be applied when, as in our dataset, there are many “undetectable or invisible” low abundant species, that prevent the attainment of a statistically accurate richness estimation [48,49]. Actually, our NGS dataset is mainly composed of a big tail of low abundance taxa known as “Rare Biosphere” (here the term “rare” is based on arbitrary cutoffs in abundance of <1% of the total community) that represent more than 96% of the microbial community found in the deep-sea sediments (Appendix A). Recent studies on microbial rare biosphere described the mechanism of “conditionally rare taxa” (CRT): microorganisms usually rare within a community that can occasionally become abundant in response to drastic environmental stress [50]. An oil spill could be a stressor, and so these microorganisms might become more abundant in the presence of high concentrations of oil [15,51,52]. Given this possibility, we explored deeper the low abundant population of our NGS dataset and found in L1 all the 6 Hydrocarbons-Degrading (HD) genera we have isolated after an enrichment with petroleum. This result supported the hypothesis that these 6 HD taxa, with the ability to degrade hydrocarbons, isolated from L1 pristine samples, could belong to a dormant pool of CRT that only bloomed in response to petroleum contamination, displaying their bioremediation potential. This interesting finding confirms the relevance of the culture-dependent approach here also applied to identify bacteria with the ability to utilize crude oil as the sole carbon source. 

After microbial characterization and isolation of bacterial strains with potential to degrade petroleum, a lab-scale bioremediation experiment was designed to investigate the potential of these strains when assembled in a consortium, to degrade petroleum hydrocarbons in a natural environment. Initially, two different carbon sources, petroleum and sodium acetate, were tested to compare their efficiency to produce high consortium biomass. Being a simple carbon source, sodium acetate is rapidly uptake by most bacteria and had been already tested in previous biodegradation experiments [37]. Our results showed that the HDB_DSS_L1 bacterial consortium grown on sodium acetate was able to survive and grow in petroleum in the bioaugmentation treatment. This finding showed the efficiency of sodium acetate as a non-hazardous alternative to petroleum to grow biomass for application in bioremediation strategies. 

Data from hydrocarbon degraders abundance, obtained by MPN, indicated that both biostimulation and bioaugmentation treatments (either with inoculum pre-grown in sodium acetate or in petroleum) can be highly efficient. Biostimulation showed an increase in the abundance of degrading bacteria between the beginning (T0) and the end (T15) of the experiment, while for bioaugmentation high abundance of degraders was observed since the beginning of the experiment due to the addition of the microbial consortia. Moreover, for each treatment, we evaluated the ability of the microbial consortium for utilizing the crude oil as carbon source through TPH analyses. We found the lowest hydrocarbons degradation in natural attenuation (18%), followed by biostimulation treatment (21%), bioaugmentation with inoculum pre-grown in petroleum treatment (23%), and bioaugmentation with inoculum pre-grown in acetate treatment (30%). The same trend for higher degradation in the BAa (bioaugmentation treatment with inoculum pre-grown in sodium acetate), when compared to BAp (bioaugmentation treatments with inoculum pre-grown petroleum), has been observed in a previous study [37], indicating that this result was associated with the higher bacterial biomass growth obtained in the presence of acetate compared with petroleum. 

In the present work, we obtained relatively low degradation rates of hydrocarbons compared with other studies that tested the hydrocarbon degradation potential of consortia obtained from coastal sediments [53,54,55]. Nevertheless, different factors that affect hydrocarbon degradation rates should be considered when making this comparison. Among them are the diversity of the experimental conditions, the different composition and concentration of hydrocarbons used in the experiments, previous exposure to hydrocarbons sources, and the origin of the microbial consortia. Tyagi et al. [12] reported that inadequate bioavailability of the hydrocarbons, due to low water solubility, can be a limiting step in biodegradation. Higher degradation could be obtained, for instance, by combining biostimulation, bioaugmentation and biosurfactant addition, as the later could help microorganisms to access hydrocarbon compounds, ensuring relatively faster degradation rates [56]. Bento et al. [54] compared natural attenuation, biostimulation and bioaugmentation, concluding that the best approach for bioremediation of diesel oil was the bioaugmentation performed by inoculating microorganisms pre-selected from a contaminated site. Indeed, prior exposure of the inoculum to the pollutant caused an adaptation of the microorganisms to utilize the hydrocarbons as carbon and energy sources, shortening the lag phase and accelerating the beginning of the exponential phase [57]. Furthermore, Potts et al. [58] in a recent work reported differences in hydrocarbons removal between shallow and deep sediment consortia (50–58% and 33–42%, respectively), highlighting the role of consortium origin. 

To the best of our knowledge, the most of the bioremediation studies on deep-sea environment investigated the hydrocarbon degradation potential of water column microbial consortia [59,60], fewer also studying deep-sea sediment consortia [25,43,61,62]. However, the latter researchers described bioremediation experiments in microcosms assembled with sterilized culture media. In this context, the present research aims to bring novelty in deep-sea oil spill bioremediation, testing a DSS_HD bacterial consortium in microcosm experiment assembled with natural seawater instead of sterilized media. This condition contributes to mimic a realistic oil spill scenario in which microbial interactions (competition or cooperation) between natural seawater community and added bacteria may occur, possibly interfering with the biodegradation process. 

The identification of the recovered bacterial strains at the end of the bioremediation experiment showed that most of the bacterial strains used to prepare the bioaugmentation consortium, inoculated in BAa and BAp, were able to survive, grow, and use petroleum as carbon source in a naturally competitive environment. This finding highlights their potential for future application in bioremediation treatments. Indeed, 6 out of the 8 strains used in the assembled HDB_L1_DSS consortium were recovered at the end of the experiments. The three genera *Achromobacter*, *Acinetobacter*, and *Pseudomonas* have been previously reported as the most important hydrocarbon-degrading taxon in soil by producing alkane hydroxylases [63]. On the contrary, there are no references relating to hydrocarbons degradation in *Microbacterium testaceum*. Yao et al. [64] isolated *M. testaceum* from an extremely phosphate-poor lake and showed that its macromolecular composition can be altered, allowing its growth in extremely phosphorus-deficient environments, revealing substrate specialization. In the present study, it was possible that isolate *M. testaceum* used crude oil as the only carbon source, bringing a new insight into this species. Two of the bacterial strains of the assembled consortia were not recovered at the end of the experiment. One of them was *Sphingobium* sp., an already known PAH-degrading genus, isolated from different marine environments including DSS, as reported in Louvado et al. [16]. The other strain was *Rheinheimera aquimaris,* which was first described as a marine bacterial strain by Yoon et al. [65]. Few reports about this strain and its characteristics have been published [66], and its features seem to have no association with the biodegradation of petroleum hydrocarbons, although the genus *Rheinheimera* is known as hydrocarbons degrader [67]. 

Apart from the bacterial strains of the assembled HDB_L1_DSS consortium, five additional strains were recovered at the end of the experiment. These may have originated from the seawater (collected at the shore) used in the bioremediation experiments. One of those strains, *Leeuwenhoekiella marinoflava*, previously classified as *Cytophaga marinoflava*, is a halotolerant marine bacterium already isolated from seawater [68]. Members of the *Cytophaga* genus have previously been described as being able to consume crude oil hydrocarbons and were detected in soils heavily contaminated with PAHs [69]. In addition, the Rhodobacteraceae family includes the *Sulfitobacter* genus that often lives in marine environments and may be involved in the biodegradation of chemically dispersed oil [45]. From the same family, the marine genus *Aquicoccus* is represented by only one species, *Aquicoccus porphyridii*. Strain BPM3, isolated during the microcosm experiment is correlated with this genus but may represent a new taxon as it has a similarity percentage lower than 97.80% with *A. porphyridii*, its closest type strain [70]. The genus *Frondibacter*, belonging to the Flavobacteriaceae family, is constituted only by two species of marine bacteria *Frondibacter aureus* and *Frondibacter mangrovi*. The BPM 4 strain is associated with this genus; however, it may also represent a new taxonomy as it only has a similarity percentage of 95.71% with the closest type strain [65]. None of the last two genera has been yet associated with hydrocarbons bioremediation. However, more studies are needed to understand whether these strains are able or not to degrade hydrocarbons or if they only tolerate this type of compounds. Furthermore, it would be important to continue studying these strains to validate their taxonomic identification, since new phylogenies favor the discovery of new metabolic pathways.

So, our bioremediation experiment clearly allowed to show the potential of natural microbial communities present in deep sea sediment for bioremediation applications, indicating that they can interact and survive in natural media contaminated with petroleum. 

## 5. Conclusions

The present study contributes to bring novelty about deep-sea oil spill bioremediation, testing a DSS_HD bacterial consortium in a microcosm experiment assembled with natural seawater and petroleum, in order to simulate an oil spill at lab-scale in a semi-realistic scenario. Bioaugmentation treatments with inoculum pre-grown either in sodium acetate or petroleum increased hydrocarbons degradation comparatively to natural attenuation, demonstrating the potential of these bioremediation strategies for the removal of hydrocarbon from contaminated sites. High throughput sequencing reveals to be a precious tool that allowed the detection of low abundant genera, i.e., microbial community with an abundance <1%, with hydrocarbon degradation potential in pristine samples. They belong to a long tail of rare taxa that may serve as a microbial seed-bank, which may increase in relative abundance and become active in response to environmental perturbations such as an oil spill. Our results contribute to highlight the bioremediation potential of these low abundance genera harbored in deep marine environments. However, further investigations on the ecological role and functional profile of the DSS rare biosphere will provide new insights into its biotechnological potential.

## Figures and Tables

**Figure 1 microorganisms-09-02389-f001:**
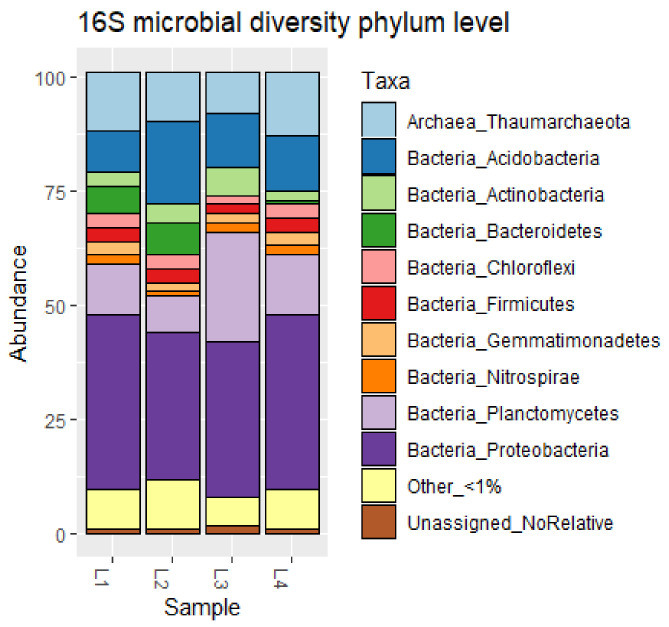
Relative abundance of prokaryotic phyla in deep-sea samples detected by NGS. Other <1% groups the phyla with abundance lower than 1%; Unassigned_No_Relative: groups the phyla without any close relatives.

**Figure 2 microorganisms-09-02389-f002:**
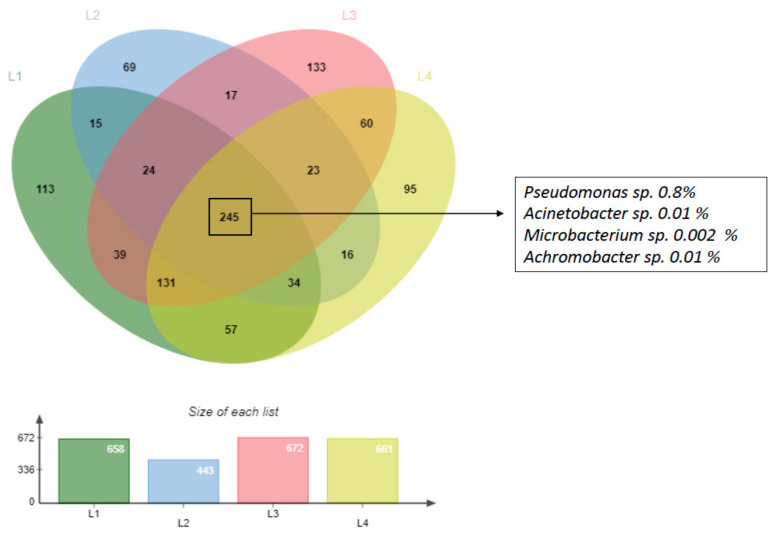
Venn diagram showing the low abundant microbial community detected in the four deep-sea sediment samples highlighting those detected in all samples. Percentage values correspond to the average abundance of the genera in the 4 samples. The bar plot represents the number of low abundance taxa within each site.

**Figure 3 microorganisms-09-02389-f003:**
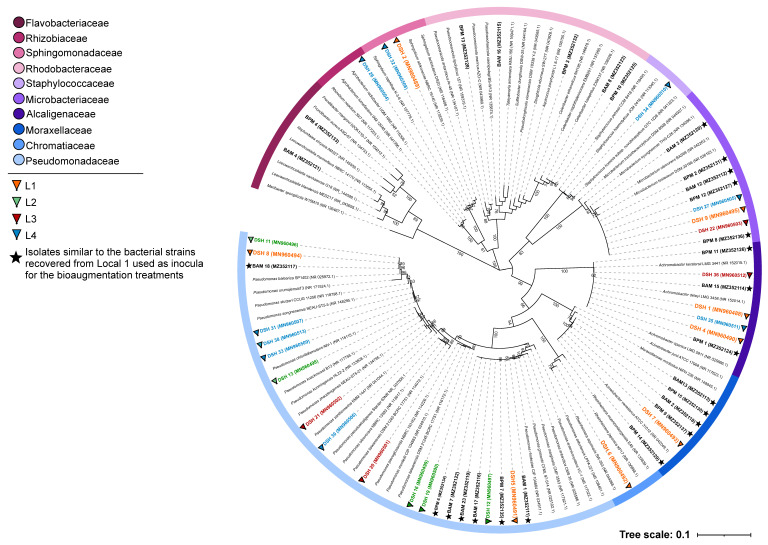
Phylogenetic tree (16S rRNA gene) of strains isolated. Maximum Likelihood analysis was performed in MEGA 7 with 53 strains and their GenBank nearest neighbours. The tree was generated using 1412 bp and 1000 bootstraps. Bootstrap values (%) are represented on tree branches (values below 60% have not been displayed). Numbers in parenthesis correspond to GenBank accession numbers.

**Figure 4 microorganisms-09-02389-f004:**
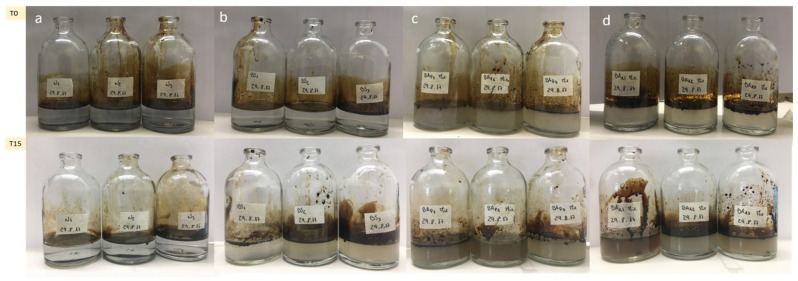
Visual inspection of oil degradation in the beginning (T0) and end (T15) of the bioremediation experiments in controlled laboratory conditions. The four treatments are natural attenuation (**a**), biostimulation (**b**), bioaugmentation with inoculum pre-grown in petroleum (**c**) and bioaugmentation with inoculum pre-grown in acetate (**d**).

**Figure 5 microorganisms-09-02389-f005:**
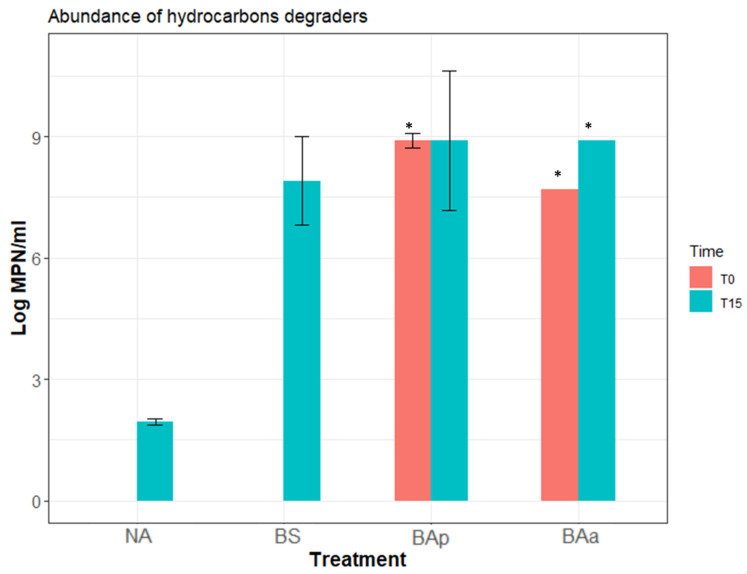
Hydrocarbon-degrading microorganisms estimated by most probable number (log_10_ MPN, mean and standard deviation, *n* = 3) of the bioremediation experiments. MPN tests were carried out at the first (T0) and last (T15) days of the experiments. *** Methodological saturation. NA: natural attenuation; BS: biostimulation; BAp: bioaugmentation with inoculum pre-grown in petroleum; BAa: bioaugmentation with inoculum pre-grown in acetate.

**Figure 6 microorganisms-09-02389-f006:**
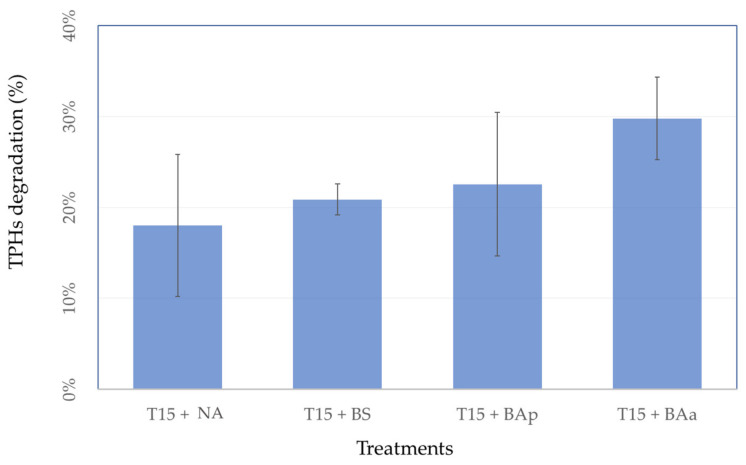
Total Petroleum Hydrocarbon degradation observed at the end of the 15-day experiment (mean and standard deviation, *n* = 3). T15 + NA: natural attenuation; T15 + BS: biostimulation; T15 + BAp: bioaugmentation with inoculum pre-grown in petroleum; T15 + BAa: bioaugmentation with inoculum pre-grown in acetate.

**Table 1 microorganisms-09-02389-t001:** Sampling points.

N	Lat (N)	Lon (W)	Depth (m)	Content
L1	33.9175	−37.5054	−1067	Sediment
L2	33.92297	−37.5103	−1072	Sediment
L3	33.92297 start 33.91748 end	−37.5103 start −37.5053 end	−1065 to −1073	Composite Sediment
L4	33.91748	−37.5053	−1067	Sediment

**Table 2 microorganisms-09-02389-t002:** Phylogenetic identification of bacterial strains recovered from all the four sample sites (L1, L2, L3, L4) and of the bacteria strains isolated at the end of the microcosms bioremediation experiment for the different treatments: bioaugmentation with inoculum pre-grown in petroleum (were named as BPM) and bioaugmentation with inoculum pre-grown in acetate (were named as BAM). DSH= Deep-Sea Hydrocarbon degraders (internal experiment code).

	Isolate	Closest Identification *	Similarity (%)	Sequence Length (bp)	GenBank Acession Number
**L1**	DSH 1	*Achromobacter* sp.	100	1384	MN960488
	DSH 2	*Sphingobium* sp.	99.71	1355	MN960489
	DSH 4	*Achromobacter* sp.	99.93	1377	MN960490
	DSH 5	*Pseudomonas* sp.	100	1386	MN960491
	DSH 6	*Rheinheimera aquimaris*	99.72	1406	MN960492
	DSH 7	*Acinetobacter venetianus*	99.86	1397	MN960493
	DSH 8	*Pseudomonas balearica*	99.64	1401	MN960494
	DSH 9	*Microbacterium testaceum*	100	1362	MN960495
**L2**	DSH 11	*Pseudomonas balearica*	99.86	1399	MN960496
	DSH 12	*Pseudomonas* sp.	100	1367	MN960497
	DSH 13	*Pseudomonas* sp.	99.71	1402	MN960498
	DSH 16	*Pseudomonas* sp.	100	1402	MN960499
	DSH 19	*Pseudomonas* sp.	99.93	1402	MN960500
**L3**	DSH 20	*Pseudomonas pseudoalcaligenes*	99.86	1402	MN960501
	DSH 21	*Pseudomonas zhaodongensis*	99.93	1400	MN960502
	DSH 22	*Microbacterium testaceum*	99.35	1378	MN960503
**L4**	DSH 26	*Agrobacterium* sp.	100	1350	MN960504
	DSH 27	*Microbacterium testaceum*	99.93	1386	MN960505
	DSH 30	*Pseudomonas* sp.	99.79	1402	MN960506
	DSH 31	*Pseudomonas* sp.	100	1400	MN960507
	DSH 32	*Sphingobium* sp.	99.78	1353	MN960508
	DSH 33	*Pseudomonas* sp.	99.86	1398	MN960509
	DSH 34	*Staphylococcus hominis subsp*. *novobiosepticus*	99.93	1422	MN960510
	DSH 35	*Achromobacter* sp.	100	1396	MN960511
	DSH 36	*Achromobacter* sp.	99.93	1384	MN960512
	DSH 38	*Pseudomonas* sp.	99.93	1402	MN960513
**Bioaugmentation With Inoculum Pre-Grown In Acetate (BAa)**	BAM 1	*Pseudomonas* sp.	99.93	1397	MZ352111
	BAM 2	*Acinetobacter* sp.	99.93	1390	MZ352118
	BAM 3	*Microbacterium testaceum*	99.86	1382	MZ352120
	BAM 4	*Leeuwenhoekiella marinoflava*	99.57	1385	MZ352121
	BAM 7	*Pseudomonas* sp.	99.93	1403	MZ352122
	BAM 8	Rhodobacteraceae	100	1323	MZ352123
	BAM 12	*Microbacterium testaceum*	99.78	1391	MZ352112
	BAM 13	*Acinetobacter* sp.	99.93	1389	MZ352113
	BAM 15	*Achromobacter* sp.	99.93	1391	MZ352114
	BAM 16	*Sulfitobacter* sp.	99.95	1331	MZ352115
	BAM 17	*Pseudomonas* sp.	99.93	1403	MZ352116
	BAM 18	*Pseudomonas balearica*	99.93	1403	MZ352117
	BAM 23	*Pseudomonas* sp.	99.93	1391	MZ352119
**Bioaugmentation With Inoculum Pre-Grown In Petroleum (BAp)**	BPM 1	*Achromobacter* sp.	100	1380	MZ352124
	BPM 2	*Microbacterium testaceum*	99.85	1377	MZ352131
	BPM 3	*Aquicoccus* sp.	96.97	1321	MZ352132
	BPM 4	*Frondibacter* sp.	95.71	1375	MZ352133
	BPM 6	*Pseudomonas* sp.	100	1394	MZ352134
	BPM 7	*Pseudomonas* sp.	100	1396	MZ352135
	BPM 8	*Microbacterium testaceum*	99.93	1377	MZ352136
	BPM 9	*Acinetobacter* sp.	99.93	1399	MZ352137
	BPM 10	Rhodobacteraceae	100	1329	MZ352125
	BPM 11	*Achromobacter* sp.	99.31	1394	MZ352126
	BPM 12	*Microbacterium testaceum*	99.93	1384	MZ352127
	BPM 13	*Pseudooceanicola marinus*	99.92	1328	MZ352128
	BPM 14	*Acinetobacter* sp.	99.93	1402	MZ352129
	BPM 15	*Acinetobacter* sp.	99.93	1403	MZ352130

* According to Nucleotide collection (nr/nt) database from NCBI BLAST.

## Data Availability

The data presented in this study are available in article.

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
