# Peer review of "Diversity and Hydrocarbon-Degrading Potential of Deep-Sea Microbial Community from the Mid-Atlantic Ridge, South of the Azores (North Atlantic Ocean)"

_microorganisms, 2021, doi:10.3390/microorganisms9112389_

Round 1

Reviewer 1 Report

  • ribosomal RNA (rRNA) gene: 2) also, culture dependent – change : for ;
  • Concentration and quality of DNA were measured by fluorometry using the Qubit fluorometric quantitation kit – how did you measure quality in Qubit? Qubit is used for quantification only.
  • were transferred to a 100-mL serum flask and used – where the experiments carried out under aerobic, microaerobic or anaerobic conditions? Please specify. Same for the cultivation of plates.
  • ® Bacterial DNA kit (Omega Bio-tek). DNA quantification was performed with the kit Q - How did you extract DNA from the isolates?
  • The PCR mixture PCR conditions – add and between mixture and PCR
  • Section 2.4 – Did you deposited the sequences? If so, state that at the end of the section.
  • Section 2.5 - where the experiments carried out under aerobic, microaerobic or anaerobic conditions? Please specify
  • the same to be used in the bioremediation experiment, to create inocula pre-grown in acetate (EA) or
    in petroleum (EP) – this sentence” is not clear to me. Could you rephrase so it is clear what you mean with “the same to be used?
  • isolated from light – change isolated for “in darkness” I haven’t seen the use of isolated from light in this context at all.
  • crude oil added on a proportion of 20:0,5 – change 0,5 for 0.5
  • a mixture of the eight bacterial strains was inoculated in 250 – here you can say “the flasks were inoculated with a mixture of eight bacterial strains” or “eight bacterial strains were added to sterile 250mL glass flasks”.
  • (A) that was added daily to the cultures at – for how long?
  • namely nitrogen (as KNO3) (40 mM) and phosphorus (as KH2PO4) (8 mM) – please, be consistent throughout the text, you are using g/L (Na-acetate) and mM. Choose one of the two forms to give concentration. I suggest using g or mg/L.
  • Section 2.5 – how much biomass did you use to inoculate the bioremediation experiments? Did you use the whole re-suspended pellet?
  • Section 2.5. – I imagine you did not inoculate the experiment N and BS, please add a short statement clarifying this point.
  • After 3–4 days of incubation at 28 C, morphologically – missing ° in 28 °C
  • Section 3.1 – please check that commas and dots in figures are used correctly: e.g. A total of 239.407 sequences of 16S rRNA gene were – is the 239.407 or 239,407 – remember that in English the comma is used for thousands (e.g. 1,000) and the dot for decimals (e.g. 0.5). If the dot is correctly used in 239.407 and the others, please give only one decimal. Same in Figure 2S and Table S1.
  • When the analysis was made at deepest taxonomic level - what do you mean with deepest taxonomic level. The correct terminology is “the lowest taxonomic level” meaning that you were able to identify to family and “higher taxonomic level” meaning that you only identify phylum.
  • The microbial community at deepest level was represented at 96% by “low abundant” taxa with a relative abundance below 1%: - not sure what you mean with deepest level. Do you mean that 96% of the community was composed by taxa with only 1% relative abundance? You need to show with a graph this finding as it is important.

The microbial community at deepest level was represented at 96% by “low abundant” taxa with a relative abundance below – change at for by and by for of and delete a.

  • The latter percentage included 4 of the 6 genera constituting Hydrocarbons-Degrading – which percentage? The 10% in parenthesis? If it is in parenthesis, it looks not important, I suggest changing “the latter percentage” for “the 245 common taxa included 4 of ….”
  • 6 genera constituting Hydrocarbons-Degrading Bacterial consortium (HDB_L1_DSS) – already shown what the abbreviation means in section 2.5, delete Hydrocarbons-Degrading Bacterial consortium and only use the abbreviation. Same in section 3.2, section 3.6.
  • Rheinheimera sp. just in L1 with 0.03% and Sphingobium sp. in L1/L3 with 0.3 % of abundance – is it relative abundance? If so change it.
  • the HDB_L1_DSS consortium were detected just in L1 or in two environmental samples – change or for and
  • one of which Pseudomonas balearica – verb missing, add were between which and Pseduomonas
  • A Rheinheimera aquimaris and Microbacterium testa­ ­– change A for two isolates identified as … Same in the next sentence
  • Among the four deep sediment locations, L1 recorded the higher bacteria diversity – are there more samples than the deep ones? you only collected 4 samples, no need to say deep sediment
  • Figure 3 – it is not possible to read the text within the figure.
  • Table 2 and Figure 3. What is local 1, local 2, etc? Is that L1, L2,etc, if so, change local 1 for L1 as it is confusing the way it is now. What does DSH stand for?
  • For this reason, L1 isolates were selected to produce the Deep-Sea – I do not see the reason why it was L1 selected, I see that you obtained several isolates from L1, but at the end, you used isolates, not an enrichment to produce the consortium. Therefore, the fact that the L1 sample had highest bacterial diversity is not a reason to select those isolates from L1 unless I am missing something that is not explained. You could have used any isolated from any other site and still obtain the same results.
  • Figure 3. What did you use as outgroup? Please, indicate.
  • Section 3.3. You already state what BS, Bap and the other acronyms are, you do not need to repeat it throughout the text. Please, correct that in the manuscript.
  • Natural attenuation was evaluated in flasks only with seawater and crude
    oil to estimate oil degradation by indigenous populations. For the other treatments tested
    (BS, BAp, BAa), nitrogen and phosphorus (40 mM and 8 mM, respectively) were added to
    avoid metabolic N and P limitations and increase the metabolic activity. For bioaugmentation, two bacterial consortia pre-grown in petroleum or sodium acetate as carbon
    sources were used as inocula – this is material and methods, move it to that section.
  • , suggesting that crude oil degradation occurred during the four days of enrichment. – you mention that the pictures where taken on the first day, and in this sentence you say four days. Change the paragraph so the timeframe give is coherent.
  • Section 3.3. and section 3.5 should be combined. The visual inspection, even relevant, is not a measurement of the efficiency of the bioremediation and should be combined with the results provided
  • Figure 5a and 5b – these two figures should have only one legend if they are a and b, otherwise, they should be named with consecutive numbers.
  • The four DSS samples showed a similar community structure, although alpha diversity indices showed higher values in the L3 sample – you mention here the alpha diversity, but I cannot see results or methodology regarding the calculations related to alpha diversity, there is only a brief mention in section 3.1 that refers to figure S2
  • to the nature of L3 composed sample that contributes to increase the number of OTUs. – change composed for composite
  • Moreover, in terms of community (OTU) richness Chao1 also showed higher - you mention here the Chao 1 index, but I cannot see results or methodology regarding the calculations related to it. There is only a brief mention in section 3.1 that refers to figure S2
  • This was observed not only by visual inspection but also by comparing results obtained – which results? Explain further.
  • Nevertheless, based on the visual inspection of the flasks’ treatments, bioaugmentation seems to be more efficient than biostimulation for biodegrading crude oil. However,
    petroleum’s degradation was noticed in all treatments with pollutant’s dispersion on the
    interface oil-seawater and/or as petroleum’s droplets deposition at the bottom of the
    – I think visual inspection should be used to determine the efficiency of the method. The results analysing the TPHs show that the BAa is more efficient degrading the PH.
  • Tendentially – never heard that word before, and it is not in the dictionary. Change it and rephrase.
  • beginning of the log phase – logarithmic/exponential phase?

Discussion

The discussion lacks discussion, it is a collection of microbial species characteristics without showing the relevance of to the work presented in the manuscript. For example, what is the importance of Thaumarchaeota in your research? What is the relevance of having higher/lower Chao index? Also, in other parts it looks it is a literature review, but there is not a discussion between the results obtained in the study and the results in the cited studies. For example in the paragraph starting as: Despite the relatively low degradation rates obtained in our experiment compared with other studies [49,55,56]… Why it is important that you use natural seawater, why the bioaumetnation, why did you obtain lower degradation rates, what is the relevance of your study compared to the others?

Which useful information can you find expanding our knowledge on the diversity of natural microbial assemblages in the DSS? I see that you obtained a new consortium, but the fact that the samples have more or less diversity is irrelevant. You could have obtained the consortium from another sample with less diversity as long as HD bacteria were present in the sample.

There is a need in expanding the discussion about the results obtained in the TPH degrading experiment, and why it is important.

Why acetate is better for growing HD bacteria?

At the moment, it seems that there are three unconnected parts in the manuscript: the metabarcoding, the  degradation of hydrocarbons and the isolation of bacteria, but the relation and importance of these three parts is not clear in the discussion.

Why did you isolated the bacteria after the hydrocarbon degrading experiment? Why it is important? Why did you not analysed the consortium at the end of the experiments? This could have given you a better understanding of the effect of nutrient addition and consortium addition on the microbial community evolution after the experiments.

Conclusions

  • The identification of the strains recovered after the biodegradation experiments contributes to elucidate the potential of these bacterial strains regarding crude oil degradation. – how does contribute to elucidate the mentioned potential?
  • Our results contribute to highlight the importance of the hydrocarbon-degrading bacteria harbored in marine cold environments like DSSs. – I don’t see how the work highlight the importance of the HD (hydrogen degrading) bacteria. What I can conclude from the work is that the addition of nutrients (N and P) is important to enhance the grow of HD bacteria and that the adaptation of the consortium to the contaminant also is important.

Author Response

Thank you.

Reviewer 2 Report

The authors present an analysis of microbial diversity of a deep-sea sediment microbial community isolated from the mid-Atlantic ocean. The data provide new information on the microbial distribution of this particular environment and suggest the potential identification of several new talons. As such, this is a valuable contribution to the literature. 
Less convincing is the study of the potential of a 'constructed' consortium, based on the diversity of the species identified, to degrade petroleum-based hydrocarbons. This was too qualitative and as an example, it was very difficult to follow the arguments and to visualize what the authors were proposing as petroleum dispersion and degradation shown in Figure 4. This was the weak aspect of this manuscript.

The manuscript was generally well written but the authors need carefully to check some minor aspects of grammar, particularly the correct use of adverbs and definite and indefinite articles.

Author Response

Thank you. 

Maria Paola Tomasino

Round 2

Reviewer 1 Report

The manuscript have improved considerably thanks to the very well written and content of the discussion. I just have some minor comments:

Line 234: concentrations are given for the nutrient or for the salt? Please, clarify.

Line 241: BS, BAa and BAp were compared with N (is this natural attenuation or nitrogen?) If you use N for natural attenuation it can be confused with nitrogen. Maybe you could use NA for natural attenuation.

Line 547: why is there more freshly material deposited in the L3. Add a small sentence about it.

Line 544: change deeper explored to explored deeper

Author Response

Thank you 

kind regards 

Maria Paola Tomasino 
